# Molecular Characterization and Epidemiology of Antibiotic Resistance Genes of β-Lactamase Producing Bacterial Pathogens Causing Septicemia from Tertiary Care Hospitals

**DOI:** 10.3390/antibiotics12030617

**Published:** 2023-03-20

**Authors:** Mohammad Riaz Khan, Sadiq Azam, Sajjad Ahmad, Qaisar Ali, Zainab Liaqat, Noor Rehman, Ibrar Khan, Metab Alharbi, Abdulrahman Alshammari

**Affiliations:** 1Centre of Biotechnology and Microbiology, University of Peshawar, Peshawar 25120, Pakistan; 2Department of Computer Science, Virginia Tech, Blacksburg, WV 24061, USA; 3Department of Health and Biological Sciences, Abasyn University, Peshawar 25000, Pakistan; 4Department of Pathology, Khyber Teaching Hospital, Peshawar 25120, Pakistan; 5Department of Pharmacology and Toxicology, College of Pharmacy, King Saud University, P.O. Box 2455, Riyadh 11451, Saudi Arabia

**Keywords:** septicemia, *Escherichia coli*, *Pseudomonas aeruginosa*, antibiotic resistance, antibiotic resistance genes, mutational analysis

## Abstract

Septicemia is a systematic inflammatory response and can be a consequence of abdominal, urinary tract and lung infections. Keeping in view the importance of Gram-negative bacteria as one of the leading causes of septicemia, the current study was designed with the aim to determine the antibiotic susceptibility pattern, the molecular basis for antibiotic resistance and the mutations in selected genes of bacterial isolates. In this study, clinical samples (n = 3389) were collected from potentially infected male (n = 1898) and female (n = 1491) patients. A total of 443 (13.07%) patients were found to be positive for bacterial growth, of whom 181 (40.8%) were Gram-positive and 262 (59.1%) were Gram-negative. The infected patients included 238 males, who made up 12.5% of the total number tested, and 205 females, who made up 13.7%. The identification of bacterial isolates revealed that 184 patients (41.5%) were infected with *Escherichia coli* and 78 (17.6%) with *Pseudomonas aeruginosa*. The clinical isolates were identified using Gram staining biochemical tests and were confirmed using polymerase chain reaction (PCR), with specific primers for *E. coli* (USP) and *P. aeruginosa* (oprL). Most of the isolates were resistant to aztreonam (ATM), cefotaxime (CTX), ampicillin (AMP) and trimethoprim/sulfamethoxazole (SXT), and were sensitive to tigecycline (TGC), meropenem (MEM) and imipenem (IPM), as revealed by high minimum inhibitory concentration (MIC) values. Among the antibiotic-resistant bacteria, 126 (28.4%) samples were positive for ESBL, 105 (23.7%) for AmpC β-lactamases and 45 (10.1%) for MBL. The sequencing and mutational analysis of antibiotic resistance genes revealed mutations in TEM, SHV and AAC genes. We conclude that antibiotic resistance is increasing; this requires the attention of health authorities and clinicians for proper management of the disease burden.

## 1. Introduction

Blood is a connective tissue which forms about 8% of total body weight; 5–7 L of blood is present in an average human body. The main components of blood are plasma (liquid portions 55%) and cells (45%): white blood cells, platelets and leucocytes [1,2]. Blood functions as the transportation medium for nutrients and aids in the excretion of waste materials by specialized organs. In vertebrates, blood is important for the maintenance of the body’s temperature [3].

Blood is a sterile medium, but its contamination with pathogens or toxins leads to blood stream infections (BSIs), which are some of the leading causes of mortality and morbidity around the globe. BSIs are associated with fatal health conditions, which require admission to intensive care units [4]. In the United States, BSIs have been correlated with various risk factors, including exposure to microorganisms and the use of central venous catheters [5]. Causative agents for septicemia vary from region to region; it can be caused by both Gram-positive and Gram-negative bacteria, the most common of these being *E. coli*, *P. aeruginosa*, *Staphylococcus aureus*, *Klebsiella pneumonia* and *Salmonella typhi* [6]. Among these, the Gram-negative bacteria most associated with septicemia is *E. coli* [7,8,9].

The most common classes of antibiotics used to treat BSIs are penicillin, cephalosporins, aminoglycosides, glycopeptides, lincosamides, tetracyclines, fluoroquinolones and carbapenems. Due to overuse and misuse of antibiotics, bacteria have developed resistance to them, resulting in global health hazards [10,11]. Drug resistance in *E. coli* and other Gram-negative bacteria continues to rise, resulting in the emergence of multidrug-resistant strains. Treating the infections caused by these pathogens is a challenging issue [12]. An estimated 700,000 patients die globally each year due to high antibiotic resistance, and this number continues to rise [13]. A study in 2017 reported a total of 48.9 million cases of morbidity and 11 million of mortality worldwide, which constitutes a total of 20% mortality. Of the total, 85% of the cases of sepsis, including those of sepsis associated with death, were reported in low-middle income countries, and in Pakistan 60% of sepsis cases were fatal because of the infection being caused by multidrug-resistant strains and the misuse of antibiotics [14,15]. A study on neonatal sepsis in Sub-Saharan Africa revealed that *E. coli* accounted for 10% of the reported cases and was mostly resistant to aminoglycosides and β-lactams [16].

Determining the common pathogens and the antimicrobial susceptibility pattern causing septicemia is essential in order to select appropriate antibiotic therapies to decrease mortality and morbidity [17]. Keeping in view the importance of Gram-negative bacteria as one of the leading causes of septicemia, the current study was designed with the aim to determine the antibiotic susceptibility pattern, the molecular basis for antibiotic resistance and the mutations in selected genes of the bacterial isolates in Peshawar, Khyber-Pakhtunkhwa, Pakistan.

## 2. Results

Out of the total blood samples (n = 3389) from males and females of various age groups, 443 (13.07%) were found to be positive for bacterial growth. A total of 238 (12.5%) positive samples were from male patients and 205 (13.7%) were from female patients. Of the 443 bacterial isolates, 59.1% (n = 262) were identified as Gram-negative. The highest number of bacterial isolates were of *E. coli*, 184 (41.5%)*,* followed by *P. aeruginosa,* 78 (17.6%). The highest ratio of *E. coli* (n = 184) was observed in the age group 41–60 years, at 50 (27.1%), followed by 21–40 years, at 48 (26.01%). Similarly, the highest ratio of *P. aeruginosa* (n = 78) was observed in patients older than 60 years, 16 (50%), followed by 41–60 years, 7 (21.8%), as mentioned in Table 1.

### 2.1. Identification of Bacterial Isolates

All the isolates were identified by being cultured on MacConkey and blood agar media, followed by Gram staining (pink color colonies under microscope), API strips (as per API codes and reading scales) and on the molecular level by USP for *E. coli* and oprL for *P. aeruginosa* (Figure 1).

### 2.2. Antibiotic Susceptibility Pattern of Clinical Isolates

The resulting antibiotic sensitivity patterns of identified *E. coli* and *P. aeruginosa* revealed resistance to AMP, SXT and CIP, and sensitivity to MEM, IPM and TOB (Table 2).

### 2.3. Determination of Minimum Inhibitory Concentration

The potency of the antibiotics depends on their minimum inhibitory concentration (MIC) values. The higher the MIC value, the less potent the antibiotic, and vice versa. The *ESBLs*, MBLs and AmpC β-lactamases producing *E. coli* and *P. aeruginosa* isolates were highly resistant to CTX and CAZ with high MIC values as well as non-β-lactam drugs. SXT, CIP, DO, CN and AK were susceptible to MEM and to TGC with low MIC values (Table 3 and Table 4).

### 2.4. Phenotypic and Genotypic Identification of β-Lactamase Producers

All the positive isolates (n = 443) were screened phenotypically and genotypically for *β*-lactamase production. Out of 443 positive samples, 126 (28.4%) were ESBL positive, 105 (23.7%) were AmpC *β*-lactamase producers and 45 (10.1%) were MBL producers (Table 5).

### 2.5. Characterization of ESBLs Gene(s), MBLs and AmpC β-Lactamase Resistance Genes

Of the total 80 phenotypically detected *E. coli* isolates for ESBL production, 74 (92.5%) were positive for one or more ESBL genes. The most common gene detected was CTX-M, 56 (70%), followed by TEM, 51 (63.7%) and SHV, 28 (35%). However, in *P. aeruginosa*, the most prevalent gene was TEM (73.9%), followed by SHV (63.0%) and CTX-M (34.7%). Among the 30 phenotypically identified MBL producers, *E. coli*, 11 (36.6%) showed the presence of targeted MBLs genes, with NDM1 being the most common, 9 (30%) isolates. Similarly, the NDM1 gene were observed in 3 (20%) clinically isolated *P. aeruginosa*. Among the AmpC *β*-lactamases in *E. coli* isolates, the highest prevalence was of AmpC gene, 57 (85%), followed by CIT gene, 11 (16.4%) and the Bla-DHA gene, 8 (11.9%). Similarly in *P. aeruginosa*, AmpC was detected in 35 (92.1%) isolates, followed by 31 (81.5%) for CIT, 17 (44.7%) for DHA and 5 (13.1%) for the ACC gene (Table 6 and Figure 2).

### 2.6. Mutational Analysis ESBLs Gene(s), MBLs and AmpC β-Lactamase Resistance Genes

The sequencing data were analyzed using various bioinformatics tools; mutations were detected in Bla-TEM, Bla-SHV, Bla-ACC, Bla-NDM1, Bla-OXA1 and Bla-AAD genes but not in CTXM, AMP, CIT and DHA (Table 7). The effects of these mutations, as predicted by I-mutant software 3.0, are presented in Table 8.

## 3. Discussion

Antibiotic resistance is a major health threat and is responsible for high morbidity and mortality around the globe. Gram-negative bacteria have developed ways to combat the available antibiotics, making bacterial infections hard to treat. In the current study, the results confirmed this phenomenon, which is affecting community health and the economy. In the current study, 41.5% prevalence of *E. coli* was reported, which is similar to other findings [14]. The positivity ratios of infection of *E coli* were 64.7% in female patients and 35.3% in male patients. In our study, 27.1% of *E. coli* isolates were reported in the age group of 41–60 years; this may be due to weakened immune systems or to frequent exposure. This was followed by 26.01% in 21–40 years, which is in contrast to the reported literature [15]. The *E. coli* isolates of this study showed resistance to AMP, CTX, CAZ, CIP and LEV and sensitivity towards SCF, CO, MEM, TGC, AK, FOS and TZP, in agreement with the literature [16]. The prevalence rate of *P. aeruginosa* in the current study is 17.6%, 23.1% of this was in female patients and 76.9% in male patients, as supported by the reported study [17]. A 2016 study conducted in Pakistan found *P. aeruginosa* in 13% of septicemia patients, 55.8% males and 44.2% females. The prevalence of *P. aeruginosa* in our study at 17.6% implies that it has increased in the last few years. This directly indicates an increase in antibiotic resistance in Pakistan [18]. In the current study, ESBL genes in *E. coli* isolates were screened, in which CTX-M was detected in 70%, TEM in 63.7% and SHV in 35%. Another reported study had lower prevalence of CTX-M (57.7%), TEM (20.3%) and SHV (15.4%) [19]. Similar to that study, in the current study, 36.6% of the MBL targeted genes were detected, in which NDM1 gene prevalence was almost 30%. An Indian study reported the same results of MBL Ec with 28% prevalence of NDM-1 [20]. In this study, AmpC β-lactamase was found in 85.0%, CIT gene in 16.4% and DHA gene in 11.9% of the total clinical isolates, which supported earlier reported studies [21,22]. The mutations in the selected gene may offer a molecular explanation for the antibiotic resistance in the isolates of the current study.

## 4. Conclusions

The findings of this study have several key implications for health policymakers, clinicians and researchers. These findings highlight the need to update infection-prevention measures to be better able to manage the diseases caused by *E. coli* and *P. aeruginosa*. The increase in antibiotic resistance is an alarming situation, and necessitates a rationalization of the treatment strategy to control BSIs. The unavailability of newer drugs, and the constant increase in antibiotic resistance have led to the use of limited drugs such as colistin by physicians. This has resulted in a condition called pan-drug resistance, necessitating the discovery of new antimicrobial drugs.

## 5. Materials and Methods

The study was conducted in the Khyber Teaching Hospital Peshawar, Hayatabad Medical Complex Peshawar and the Center of Biotechnology and Microbiology, University of Peshawar, using standard microbiological procedures. A total of 3389 blood samples were collected from suspected septicemic patients in EDTA tubes aseptically, from both sexes and from various age groups, and were processed by automated blood culture systems. An overview of the whole methodology is represented in Figure 1. Informed consent was obtained from all patients on a prescribed proforma before taking blood samples.

### 5.1. Isolation and Identification

The samples were cultured on MacConkey (Merck, Rahway, NJ, USA) and blood agar (Merck, Rahway, NJ, USA) media followed by incubation at 37 °C for 24 h [23]. The isolates were identified using Gram staining (Merck, Rahway, NJ, USA) and biochemically by API kits (Biomerieux, Marcy-Etoile France) [24,25].

### 5.2. Molecular-Level Identification

For molecular-level identification of the bacterial isolates (USP for *E. coli* and oprL for *P. aeruginosa*) and detection of antibiotic-resistant genes (Table 9), DNA of the bacterial isolates was extracted using Thermo Scientific GeneJET Genomic DNA purification kits as per the manufacturer’s protocol. The extracted DNA was confirmed by gel electrophoresis (1% agarose gel in 1X triacetate EDTA buffer) and visualized by a gel documentation system.

### 5.3. Antimicrobial Susceptibility Testing

The Kirby–Bauer disk diffusion method [21] was used to determine the antibiotic sensitivity pattern of the identified bacterial isolates against selected antibiotic disks (Table 10) as per Clinical Laboratory and Standard Institute (CLSI) guidelines. The pure cultures of the bacterial isolates (0.5 McFarland standard) were inoculated on sterile Muller–Hinton agar (MHA) media, and the antibiotic disks were applied, followed by 24 h of incubation at 37 °C. The zones of inhibition were measured and were evaluated as resistant (R), intermediate (I) and sensitive (S), as per CLSI guidelines [19].

### 5.4. Minimum Inhibitory Concentration

The minimum inhibitory concentrations (MICs) of the selected antibiotics were determined using MIC strips (Table 11). The strips were placed along with inoculation of the pure isolates on sterilized MHA media, followed by overnight incubation at 37 °C [24].

### 5.5. Detection of Antibiotic-Resistant Genes by Polymerase Chain Reaction

The selected antibiotic resistance genes, as per antibiotic resistance pattern, were amplified by polymerase chain reaction (PCR) using specific primers (Table 9). The PCR contained 12.5 µL of Taq Master Mix (Thermo Fisher Scientific^TM,^, Waltham, MA, USA), 11.5 µL nuclease-free water, 0.5 µL of forward and reverse primers (oligo nucleotide Microgen, Seoul, Korea) each and 2 µL of DNA sample. Under optimized conditions (Table 9), the selected genes were amplified, run on gel electrophoresis and visualized using a gel documentation system.

### 5.6. DNA Sequencing and Mutational Analysis

The amplified PCR products of antibiotic-resistant genes were purified using a purification kit (Thermo Scientific™ GeneJET PCR Purification Kit, Waltham, MA, USA) and sequenced at Rehman Medical Institute (RMI), Peshawar, Pakistan. The FASTA sequences of the selected genes were retrieved from the GenBank–National Center for Biotechnology Information (NCBI) database after sequencing. Basic Local Alignment Search Tool (BLAST) and BioEdit 7.2 software were used to compare the FASTA sequences of the selected genes to confirm their presence in bacterial isolates and their mutational analysis [24]. The data were further analyzed for non-synonymous mutations, and I-Mutant software was used to predict the pathogenic effects of the identified mutations [25].

### 5.7. Statistical Analysis

A chi-square analysis was conducted using SPSS version 20 to find the association between the expected value of *E. coli* and the observed *p* ≤ 0.05. The number of samples (n) was set at 150 and the degree of freedom was taken at n-1. For comparative analysis, one-way analysis of variance (ANOVA) was performed among the continuous values of antibiotics with *E. coli*, and *p* ≤ 0.05 values were considered statistically significant.

## Data Availability

All the data analyzed or generated in the study are provided in the manuscript to the best understanding of the authors.

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
