# Peer review of "Molecular Characterization and Epidemiology of Antibiotic Resistance Genes of β-Lactamase Producing Bacterial Pathogens Causing Septicemia from Tertiary Care Hospitals"

_antibiotics, 2023, doi:10.3390/antibiotics12030617_

Round 1

Reviewer 1 Report

The manuscript of Mohammad Riaz Khan and co-authors (Molecular Characterization and Epidemiology of Antibiotic Resistance Genes Of Β-2 Lactamase Producing Bacterial Pathogens Causing Septicemia from Tertiary Care 3 Hospitals of District Peshawar) deals with the evaluation of the occurrence of E. coli and P. aeruginosa in blood samples of hospitalized patients in Pakistan. 3389 clinical samples were investigated in this study and show an increase of antibiotic resistant occurrence of these microbes.

However, there are some inconsistencies in this study.

Point 1: Throughout the complete text, none of the used abbreviations was explained. The first mention can be found in Table 10, which is more than annoying. Please change it. Point 2: lane 27: specific primers (E. coli (USP): as mentioned in the reference, this includes 3 different genes. Which one was analysed? Point 3: Scheme 1, page 5: as the experimental setup is explained in the text, this scheme can be deleted. Point 4: Table 1: indicate in the legend that numbers in parentheses are the frequency. Point 5: Table 1: as a total 184 isolates of E. coli are mentioned. What does the frequency of 41.5 means? Shouldn´t it be 100%? Same for the values of P. aeruginosa. Point 6: paragraph 2.1.: unclear what the authors mean: classification of E. coli or pathogenic E. coli? Point 7: indicate the marker sizes on the (at least) left side of Figure 1 and avoid positioning them into the figure, don´t use coloured arrows. Point 8: Figure 2 can be deleted without loss of any significant information. The data are provided by the tables and the figure does not shown a representative overview of the results. Point 9: what are the consequences of the deletion of nucleotides within the gene? Are these strains sensitive against the antibiotic? Please include this in table 8. Point 10: The first two rows of table 11 should be change. MIC Strips are not mentioned in the text; therefore, it is more reasonable to start with the second row as the symbols are mentioned. Point 11: lanes 207 to 212: Within the text, nothing is indicated that only parts of the resistance genes were amplified, sequenced and used for the occurrence of mutations. This should be included in the text. The authors used the Taq polymerase for amplification of the PCR fragments. Was the error frequency of this polymerase taken into account in the sequence evaluation? If not, this should be included or a proof reading polymerase should be used. In general:

The use of upper and lower case within the text should be checked carefully

Author Response

REPLY TO THE COMMENTS

The authors are very thankful to the reviewers for their valuable comments on our manuscript and we are sure that it will polish our manuscript.  Following are para wise reply to the comments of worthy reviewers and the changes has been highlighted in the revised manuscript. 

Reviewer # 1

Point 1: Throughout the complete text, none of the used abbreviations was explained. The first mention can be found in Table 10, which is more than annoying. Please change it. 

Answer: The abbreviation has been explained where they first appeared in the text.

Point 2: lane 27: specific primers (E. coli (USP): as mentioned in the reference, this includes 3 different genes. Which one was analysed? 

Answer: USP gene is generally used as a marker gene for the identification of E. coli and this was used in the current study for the molecular-level identification of E. coli.

Point 3: Scheme 1, page 5: as the experimental setup is explained in the text, this scheme can be deleted. 

Answer: Scheme 1 was added with the purpose to summarize the whole methodology in a single scheme that can be eye-catching for the reader.

Point 4: Table 1: indicate in the legend that numbers in parentheses are the frequency.

Answer: The suggested changes have been made in the revised version of the manuscript  

Point 5: Table 1: as a total 184 isolates of E. coli are mentioned. What does the frequency of 41.5 means? Shouldn´t it be 100%? Same for the values of P. aeruginosa. 

Answer: This was a calculation mistake and has been corrected in the revised version of the manuscript

Point 6: paragraph 2.1.: unclear what the authors mean: classification of E. coli or pathogenic E. coli

Answer: The authors were unable to find this term but the E. coli in this manuscript means pathogenic E. coli

Point 7: indicate the marker sizes on the (at least) left side of Figure 1 and avoid positioning them into the figure, don´t use coloured arrows. 

Answer: The suggested changes has been made in Figure 1

Point 8: Figure 2 can be deleted without loss of any significant information. The data are provided by the tables and the figure does not shown a representative overview of the results. 

Answer: As the tables are not showing the size of the band as well as the gel images, so figure 2 is the showing the representative gel images of the selected genes

Point 9: what are the consequences of the deletion of nucleotides within the gene? Are these strains sensitive against the antibiotic? Please include this in table 8.

Answer: The I-Mutant software under the “I-Mutant prediction effect” gives the values as “increase” and “decrease” in the context of the stability of the protein after mutation and has been presented in table 8.  

Point 10: The first two rows of table 11 should be change. MIC Strips are not mentioned in the text; therefore, it is more reasonable to start with the second row as the symbols are mentioned. 

Answer: The MIC strips have been changed to E-strips

Point 11: lanes 207 to 212: Within the text, nothing is indicated that only parts of the resistance genes were amplified, sequenced and used for the occurrence of mutations. This should be included in the text. The authors used the Taq polymerase for amplification of the PCR fragments. Was the error frequency of this polymerase taken into account in the sequence evaluation? If not, this should be included or a proof reading polymerase should be used. In general:

Answer: In the current study, full length genes were amplified as we were interested in the mutational studies and amplifying part of the gene was not satisfying the objective of the study. As the length of the selected genes was less than 1 kb and the financial constraints of our laboratory, we used Taq polymerase in the study.    

Point 12: The use of upper and lower case within the text should be checked carefully.

Answer: The manuscript has been thoroughly checked for upper and lower case within the text

Reviewer 2 Report

This study aimed to determine the antibiotic susceptibility pattern, the molecular basis for antibiotic resistance, and mutations in selected genes of the bacterial isolates in Peshawar, Khyber-Pakhtunkhwa, Pakistan. Despite the important clinical implications of this study, the authors must address some areas of concern.

Areas of concern:

General

There is a need for English language editing. There are also many spacing errors throughout the manuscript.

Abstract

 This section does not mention the objective of the study.

Line 22: abdomen, urinary tract, and lungs are body organs and not infections except you meant ‘‘abdomen, urinary tract, and lung infections’’. Which clinical samples do you refer to? Be specific.

Introduction

Lines 37: check spacing here: 5-7L. Add ‘’body’’ after ‘’normal human’’

Lines 51-52: a verb is missing in the following phrase’’ hence treating the infections a challenging issue’’

Lines 56-57: This section should come before line 54.
In line 55, the authors gave the prevalence of sepsis in the Middle East and in Pakistan but failed to mention any on the antibiotic resistance problem in this part of the world given the fact the study is carried out in Pakistan.

Results

Lines 62-64 take this to the conclusion section.

Lines 66-74 take this either to the results or methodology section.

Line 80-81: check spacing between numbers and ‘’yrs’’.

Table 1: except for the first column, the contents of other columns should be centered (Idem for all the Tables except Table 5). Equally, the age groups should be reorganized because the gap between them is too wide except for the age groups 00-10 and 11-20. In fact, the immune system of someone who is 21 years and that of one of 40 years is very different. I, therefore, suggest the following age groups: 00-10, 11-20, 21-30, 31-40, 41-50, 51-60, and ˃ 60.

Section 2.5 (line 113): check the spacing between numbers and brackets

Line 116: replace ‘’similarly’’ with ‘’on the contrary’’

Line 138: replace ‘’and’’ between ‘’Bla-AAD genes’’ and ‘’no mutation’’ with ‘’but’’

Lines 144-147: remove section 2.7 because it is a repetition of section 4.7

Discussion

Generally, the authors failed to bring meaning out the findings of the result. For example, the fact that the highest prevalence of Ps.. aeruginosa in the age group ˃ 60 may confirm the opportunistic nature of this organism correlating with the weak immune system of the elderly (˃ 60).

Lines 150: Add ‘’bacterial’’ to ‘’infections’’

Line 153: write ‘’ratios’’ instead of ‘’ratio’’ and remove ‘’found’’, replace ‘’cases’’ with ‘’isolates’’

Lines 157-158: A word is missing in this statement, probably the word ’’study’’ after ‘’A’’

Conclusion

This study does not have a conclusion that should portray both the limitations and strengths of this study with possible clinical implications and research gaps as perspectives.

Materials and methods

Line 174: How was the blood collected, transported, and stored before lab analysis?

Line 177: specify the name of the manufacturers of culture media and the country of origin. Idem for all other materials such as the DNA extraction kits and PCR reagents as well as the sequencing reagents.

Lines 188-189: reference the Kirby Bauer disc diffusion method.

References

Check references 7, 8,9,12, and 18 for consistency..

Author Response

REPLY TO THE COMMENTS

The authors are very thankful to the reviewers for their valuable comments on our manuscript and we are sure that it will polish our manuscript.  Following are para wise reply to the comments of worthy reviewers and the changes has been highlighted in the revised manuscript. 

Reviewer # 2

Comments and Suggestions for Authors

This study aimed to determine the antibiotic susceptibility pattern, the molecular basis for antibiotic resistance, and mutations in selected genes of the bacterial isolates in Peshawar, Khyber-Pakhtunkhwa, Pakistan. Despite the important clinical implications of this study, the authors must address some areas of concern.

Areas of concern:

General

Query: There is a need for English language editing. There are also many spacing errors throughout the manuscript.

Answer: The manuscript has been proofread by a faculty member of the Department of English, University of Peshawar to remove any English errors. The spacing errors had been corrected throughout the manuscript.

Abstract

Query: This section does not mention the objective of the study.

Answer: The objective of the study has been added in this section

Line 22: abdomen, urinary tract, and lungs are body organs and not infections except you meant ‘‘abdomen, urinary tract, and lung infections’’. Which clinical samples do you refer to? Be specific.

Answer: The correction has been made in the revised version of the manuscript

Introduction

Query: Lines 37: check spacing here: 5-7L. Add ‘’body’’ after ‘’normal human’’

Answer: The correction has been made in the revised version of the manuscript

Query: Lines 51-52: a verb is missing in the following phrase’’ hence treating the infections a challenging issue’’

Answer: The correction has been made in the revised version of the manuscript

Query: Lines 56-57: This section should come before line 54.
In line 55, the authors gave the prevalence of sepsis in the Middle East and in Pakistan but failed to mention any on the antibiotic resistance problem in this part of the world given the fact the study is carried out in Pakistan.

Answer: The problem of antibiotic resistance has been highlighted in this part of the manuscript.

Results

Query: Lines 62-64 take this to the conclusion section.

Answer: Line 62-64 has been shifted to the conclusion section

Query: Lines 66-74 take this either to the results or methodology section.

Answer: It has been shifted to Methodology

Query: Line 80-81: check spacing between numbers and ‘’yrs’’.

Answer: It has been corrected as suggested by the worthy reviewer

Query: Table 1: except for the first column, the contents of other columns should be centered (Idem for all the Tables except Table 5). Equally, the age groups should be reorganized because the gap between them is too wide except for the age groups 00-10 and 11-20. In fact, the immune system of someone who is 21 years and that of one of 40 years is very different. I, therefore, suggest the following age groups: 00-10, 11-20, 21-30, 31-40, 41-50, 51-60, and Ëƒ 60.

Answer: It has been corrected as suggested by the worthy reviewer. The age groups were made in consensus with the physicians who helped in the collection of the samples 

Query: Section 2.5 (line 113): check the spacing between numbers and brackets

Answer: The correction has been made as suggested

Query: Line 116: replace ‘’similarly’’ with ‘’on the contrary’’

Answer: The correction has been made as suggested

Query: Line 138: replace ‘’and’’ between ‘’Bla-AAD genes’’ and ‘’no mutation’’ with ‘’but’’

Answer: The correction has been made as suggested

Query: Lines 144-147: remove section 2.7 because it is a repetition of section 4.7

Answer: Section 2.7 has been removed as suggested by the reviewer

Discussion

Query: Generally, the authors failed to bring meaning out the findings of the result. For example, the fact that the highest prevalence of Ps.. aeruginosa in the age group Ëƒ 60 may confirm the opportunistic nature of this organism correlating with the weak immune system of the elderly (˃ 60).

Answer: The discussion has been modified to highlight the importance of the findings.

Query: Lines 150: Add ‘’bacterial’’ to ‘’infections’’

Answer: Line 150, infections has been changed to bacterial infections

Query: Line 153: write ‘’ratios’’ instead of ‘’ratio’’ and remove ‘’found’’, replace ‘’cases’’ with ‘’isolates’’

Answer: The correction has been made as suggested

Query: Lines 157-158: A word is missing in this statement, probably the word ’’study’’ after ‘’A’’

Answer: The word “study” has been added after “A”

Conclusion

Query: This study does not have a conclusion that should portray both the limitations and strengths of this study with possible clinical implications and research gaps as perspectives.

Answer: A conclusion section has been added as suggested by the reviewer

Materials and methods

Query: Line 174: How was the blood collected, transported, and stored before lab analysis?

Answer: The blood was collected in EDTA tubes aseptically and directly analyzed on an automated blood culture system.

Query: Line 177: specify the name of the manufacturers of culture media and the country of origin. Idem for all other materials such as the DNA extraction kits and PCR reagents as well as the sequencing reagents.

Answer: The name of the manufacturer of culture media and the country of origin has been added

Query: Lines 188-189: reference the Kirby Bauer disc diffusion method.

Answer: A reference has been added for the Kirby Bauer disc diffusion method

References

Query: Check references 7, 8,9,12, and 18 for consistency.

Answer: Reference 7, 8,9,12, and 18 has been corrected as per the journal’s format

Round 2

Reviewer 2 Report

This study was aimed to determine the antibiotic susceptibility pattern, the molecular basis for antibiotic resistance and mutations in selected genes of the bacterial isolates in the Peshawar, Khyber-Pakhtunkhwa, Pakistan. The quality of the manuscript has been greatly improved, however the authors failed to address some few concerns raised in my previous comments.

Areas of concern:

Introduction

Lines 61-62: You cannot talk about the prevalence of E. coli, Klebsiella, Enterobacter and Pseudomonas species in Africa without previously saying their prevalence in Pakistan where this study was carried out.

Materials and methods

Line 166: Scheme 1 should not be hanging between the conclusion section and the Materials and Methods section. Please, bring it somewhere in the Materials and Methods Section.

Line 187: Indicate the concentration of the bacterial suspension used for inoculation and how did you measure it?

NB: You did not mention anywhere how ethical considerations were addressed in this study knowing that blood samples were collected from patients. Did you previously apply for ethical clearance from the hospital or any other ethical review board?

Author Response

Response to Reviewer Comments

We thank the Referee for spending time and interest in our work and for helpful comments that will greatly improve the manuscript. We have checked all the general and specific comments provided by the Referee and have made all the necessary changes according to his indications. Please refer to yellow highlighted sections in the revised manuscript

Reviewer # 2

Comments and Suggestions for Authors

This study was aimed to determine the antibiotic susceptibility pattern, the molecular basis for antibiotic resistance and mutations in selected genes of the bacterial isolates in the Peshawar, Khyber-Pakhtunkhwa, Pakistan. The quality of the manuscript has been greatly improved, however the authors failed to address some few concerns raised in my previous comments.

Areas of concern:

Introduction

Query: Lines 61-62: You cannot talk about the prevalence of E. coliKlebsiellaEnterobacter and Pseudomonas species in Africa without previously saying their prevalence in Pakistan where this study was carried out.

Answer: Line 61-62 now only focuses on E. coli as the main focus of the study was E. coli and the remaining organisms have been removed

Materials and methods

Query: Line 166: Scheme 1 should not be hanging between the conclusion section and the Materials and Methods section. Please, bring it somewhere in the Materials and Methods Section.

Answer: Scheme 1 has been placed in the Materials and Methods Section.

Query: Line 187: Indicate the concentration of the bacterial suspension used for inoculation and how did you measure it?

Answer: In the preparation of the inoculum, the resulting suspension was compared to the McFarland standard. If the bacterial suspension appears lighter than the 0.5 McFarland standard, more culture was added to the tube from the culture plate. If the suspension appeared dense than the 0.5 McFarland standard, additional saline was added to the inoculum tube in order to dilute the suspension to the appropriate density. A 0.5 McFarland standard suspension was used in the study and has been added to Line 188  

Query: NB: You did not mention anywhere how ethical considerations were addressed in this study knowing that blood samples were collected from patients. Did you previously apply for ethical clearance from the hospital or any other ethical review board?

Answer: Informed consent was taken from all the patients on a prescribed proforma before taking blood samples 

Round 3

Reviewer 2 Report

This study was aimed to determine the antibiotic susceptibility pattern, the molecular basis for antibiotic resistance and mutations in selected genes of the bacterial isolates in the Peshawar, Khyber-Pakhtunkhwa, Pakistan. The authors have addressed all the concerns raised in my previous comments though I am not completely satisfied with the issue of the ethical considerations.

Materials and methods

Ethical considerations

Ideally, an ethical clearance from an ethical institutional review board is what is needed and informed consent from patients to address ethical considerations for this type of study.